# Balancing the Bar–Influence of Social Behaviour on Sport Climbing Performance

**DOI:** 10.3390/ijerph19159703

**Published:** 2022-08-06

**Authors:** Floyd Simen, Andreas Hohmann, Maximilian Siener

**Affiliations:** Department of Training Science & Kinesiology, BaySpo–Bayreuth Center of Sport Science, University of Bayreuth, 95440 Bayreuth, Germany

**Keywords:** climbing, behaviour, performance, sport, elite, qualitative study, interview

## Abstract

**Background:** For high performance in sport climbing, various factors must be taken into account, however, social interactions during climbing sessions have not yet been considered. **Methods:** For this study, four assumptions were made: (1) elite climbers share similar patterns of behaviour; (2) these behaviour patterns influence the climbing performance in a positive way; (3) the behaviour patterns had to be formed over time, and the process of changing behaviour was catalysed by formative experiences; and, (4) the social behaviour of elite climbers shows a tendency to be on their own rather than in the centre of social events, while their behavioural roots date back to their school life. Six male higher-elite-level climbers participated in semi-structured in-depth interviews. **Results:** In school, all participants perceived themselves as not being at the centre of social interactions. Moreover, all participants described a development from egoistic behaviour towards more supportive behaviour. Two participants were able to outline specific, formative experiences (crucibles), that drastically shaped their behaviour. **Conclusion:** All participants performed best in the absence of social tensions and when the atmosphere between the climbing partners was positive. Positive atmosphere was achieved by finding the optimal personal balance between supportive and egoistic behaviour.

## 1. Introduction

Climbing has grown exponentially, and has become an exceedingly popular sport [1,2]. Climbing recently had its Olympic debut in Tokyo, records keep being shattered, and modern elite training facilities are emerging all over the place. With climbing’s growing popularity, scientific research to improve climbing performance has also advanced, even as science tries to keep up with climbing’s rapid expansion [2]. Moreover, the demands placed on athletes by the various climbing disciplines are becoming increasingly divergent. The disciplines of sport climbing are speed climbing, bouldering and lead climbing. Each discipline is also represented by international competitions. In speed climbing, the athlete attempts to complete an internationally standardised artificial route in the shortest possible time. The climber is secured with a rope from an automatic belay device at the top of the route. Bouldering is a facete of climbing, where the climber attempts to complete a climb, with suitable protection by mats (crash pads) laying on the ground. The safety can be increased further when another, trained, person is present, who guides the climber onto the pad during their fall [3]. In lead climbing, the climber attempts a higher route, where the protection provided by crash pads is considered insufficient. In this case, the climber is protected by a rope that connects the climber to a climbing partner. The climbing partner must secure the rope using an appropriate technique, and acts as counterweight in case of a fall.

Usually, sport climbing is conducted with climbing partners. This always involves social behaviour and interactions. By consensus, social behaviour is defined as “all behaviour that influences, or is influenced by, other members of the same species” [4]. Normally, only one climber boards the wall and starts climbing, and for the duration in which he or she is climbing or figuring out their moves, that line is blocked for any other climber, by the climber themselves. Furthermore, the person who belays is not able to climb another line at the same time. Therefore, ideally, every climber present should receive a fair amount of time to climb on the line they opted.

On the one hand, selfish climbers who insist on trying a line several times in a row or staying on the wall for a long time can extend their training time through this behaviour, thus gaining more experience and also increasing their training success. For this reason, it can be assumed, that professional climbers tend to act more selfishly than less successful climbers. On the other hand, this behaviour puts a strain on social relationships among climbers, which can have a direct impact not only on the enjoyment of climbing, but also on concentration and mental performance [5].

In competitions, the setting is slightly different. The competitor usually does not decide who is going to belay them. Therefore, no communication between climber and belayer is needed. In most cases, it is also predetermined when to climb, and on which route. Therefore, in most competitions, the social interaction remains between the competitor and other competitors, coaches, and spectators. However, it is questionable, so far, as to whether athletes behave differently during competition, than in their training groups.

To date, scientific and common literature has focused on the physical, mental, tactical, and technical aspects of sport climbing [2,3,6,7,8,9,10,11,12]. Additionally, injury prevention has been the focus of increasing attention, since the number of climbing-specific injuries have surged [13]. Surprisingly, the aspect of social behaviour has not received any remarkable attention in the context of performance-enhancement in climbing.

The purpose of this study is to fill this gap, by presenting first directions for the novel discussion of social behaviour in sport climbing. This will benefit the understanding of performance improvement in climbing, in general, and will extend the known palette of fields for coaches and athletes to work on, specifically. 

Four assumptions were made: (1) elite climbers share similar patterns of behaviour; (2) these behaviour patterns influence climbing performance in a positive way; (3) the behaviour patterns were formed over time, and the process of changing behaviour was catalysed by formative experiences; and, (4) the social behaviour of elite climbers shows a tendency to be on their own rather than in the centre of social events, while their behavioural roots date back to their school life. The fourth assumption was based on the work of Brandauer [14] and Saul et al. [15], where male climbers were identified as showing a tendency towards introverted behaviour. To investigate these assumptions, six higher-elite-level climbers were interviewed using an explorative, semi-structured in-depth approach, where the participants outlined their essential experiences in the context of climbing. The study includes the participants’ shared experiences from all facets of aforementioned climbing disciplines and scenarios, while focusing on outdoor lead climbing and outdoor bouldering, since this is where all participants had their personally most meaningful experiences.

It is noted that this study focuses on social behaviour within the climbing session and does not include the climber’s behaviour on social media or public events, such as TV-shows or movies.

## 2. Materials and Methods

### 2.1. Participant Selection

A pre-defined number of six athletes that met the criteria for the category of higher-elite-level climbers as defined by IRCRA (International Rock Climbing Research Association) [16] participated in this study. For males, these criteria were: lead-climbing at least 8c+ (French grade) or bouldering at least 8b (font grade); and for females the criteria were: lead-climbing at least 8c (French grade) and bouldering at least 8a+ (font grade) [16]. Commonly, lead climbing is considered as the most highly valued discipline [17]. For better comparability, all six participants were the same sex. Additionally, all were of German nationality, for better reachability and to avoid language barriers. These selection criteria were due to the study’s novel nature. It was necessary to have a sample of high performing athletes, to ensure rich experience in improving their own performance. This performance improvement was the topic they were required to reflect on in the context of social behaviour. Six athletes were identified as being the most suitable prospective participants, who most fitted the combination of applicability to IRCRA criteria, diversity of background and reachability, and were contacted. All six athletes agreed to participate; however, one of these athletes noted his unreliable reachability during the phase of data collection, therefore, the next most fitting participant was contacted, who agreed to participate in this study.

All of the six elite athlete participants are publicly renown. This implies that substantial information about the athletes is publicly accessible, which makes it easy for anyone, especially for climbing scene insiders, to identify them. Thus, according to Conzelmann [18], problematic, deeply personal information that could potentially harm the image of the participants, was not displayed, and all participants gave their explicit written permission for publishing this information. For this study, deeply personal experiences have been anonymised. Every citation and personal information that was integrated in this article, even though it was anonymised, was sent to the corresponding participant in advance for revision.

### 2.2. Interviewer

All semi-structured interviews were conducted by the lead author, F.S. At the time of the interviews, he was 22 years of age, had been climbing for ten years, competed at national level, and was regarded as an elite-level-climber according to the IRCRA scale. He had also been training youth-elite climbing groups as a licenced coach for 5 years.

### 2.3. Data Collection

The interviews followed a semi-structured concept [19]. In this narrative, explorative style, the interviewer navigated the interview through a set of open question topics, that were defined prior to all interviews (Appendix A). The interview was held in German language. The conversations lasted approximately forty-five minutes each, and were conducted via video conference. The participant and the interviewer were the only persons present.

After querying the standard profiling characteristics [16], the interview continued with a wide and open question, and then narrowed into more specific topics. The standard profiling characteristics as defined by the IRCRA [16] were: age, years of climbing experience, experience in competition, main discipline, distribution of disciplines, distribution of indoor and outdoor climbing, time spent training or climbing on average per week, and their grade-wise biggest achievement in lead climbing and bouldering. The open question was: “How did you become the climber that you are now?” Its aim was to discover the factors influencing the development of the person as a climber, and to gain a first insight into the life of the participant. After that, the interviewer navigated through the different topics in varying order. One topic was the influence on climbing performance of behaviour towards other persons. This field was chosen to discover patterns of behaviour prior to, after, and especially while being at the outdoor climbing site, in gyms or at competitions. Of specific interest was the social role the climber perceived himself to embody, which was presumed to be with a tendency towards egoistic or supportive. The participants were asked about their relationships with classmates, friends, and teammates; and how the atmosphere was, either supportive or more egoistic. Additionally, the engagement of parents was of interest, and whether it played a role in the development of the climber. How the participants behaved at competitions and how they perceived the behaviour of their competitors, was also asked. The fourth subject of interest was the presence of crucibles in the lives of the participants. “A crucible is, by definition, a transformative experience through which an individual comes to a new or an altered sense of identity” [20]. Such experiences were assumed to play a significant role in the formation of the participants’ current behaviour. Having a mentor as a guide for behaviour can also be considered as a crucible [20]. The fifth, and always last, part of the guideline was of a commentary nature, where the participant could add anything he wanted regarding the topic of behaviour in the context of climbing.

### 2.4. Analysis

The interviews were video recorded after the participants gave their explicit permission. Following each interview, the audiotapes were transcribed verbatim by the author. The transcripts were coded by the author into six head codes using the software MAXQDR to identify similarities and patterns that could be derived from the data. These head codes were: (A) “IRCRA”: used to gather the data necessary to compare this study’s participants with the participants of other studies. (B) “Factors”: the factors that most influenced the climber’s development and behaviour. (1) “Behaviour”: various examples of behaviour in the context of climbing. (2) “Behaviour on performance”: specific examples of when social behaviour was perceived as a direct influence on climbing performance. (3) “Crucibles”: examples of experiences that were perceived as drastically formative for their social behaviour. (4) “Social role”: the social role the participants perceived themselves in, especially in school. Codes (1) to (4) match the original assumptions. Codes (A) and (B) are topics of information, to help understand the behaviour in the context of the climber’s development.

## 3. Results

### 3.1. Participant Description

All participants were within the range of higher-elite-level climbers according to the IRCRA scale. In the following section, each participant is individually presented.

**Alexander Megos**, one of the first two German climbers who qualified to represent Germany at the 2021 Olympic Games in Tokyo, is considered to be one of the best climbers of all time [2]. He grew up in the Franconian region in southeastern Germany, where he started climbing with his parents at the very young age of seven. He quickly became one of Germany’s best competition and outdoor climbers. At age 18, he stopped competing at official international events, and focused on outdoor climbing and bouldering, where he became highly successful. During this time he still competed at informal, invitational competitions such as the “la sportiva legends”. To qualify and prepare for the Olympic Games, he returned to competing on official international stages in 2017. At the Olympic Games, he ultimately reached ninth place, missing finals by just one rank.

**Christian Münch** grew up in the area around Munich, and has national competition experience, even though he focuses on outdoor climbing. Grade-wise, his climbing achievements lie within the range of a higher-elite-level climber, with a current level of 9a in lead and 8b in bouldering.

**Moritz Welt** currently ranks in ninth place worldwide, and in first place nationally, on the lead climbing ranking of website 8a.nu. On this platform, a ranking is created upon self-recorded ascents on rock. Certainly not every climber logs their ascents on this platform, however, it is still a strong indicator of the level of fitness on rock. Mainly an outdoor lead climber, Moritz has lead climbed 9a+, and has bouldered 8c.

**Philipp Hrozek** was one of the strongest outdoor boulderers in the Franconian region and in the area around Munich. He also competed at a national level for a short time, but felt that this style of climbing did not suit him. Mostly, he climbed and trained with his brother. In 2016 he had a severe accident, which compromised his ability to generate force by will. Nonetheless, he managed to perform a comeback in paraclimbing, at which he now competes at an international level, achieving second place in 2018 in a para-lead-climbing World Cup, as his best result so far.

**Stefan Vogt** grew up around Berlin. One of his personal highlights was the ascent of “Action Direct” which was the first lead route in the world to have been graded 9a. Additionally, in bouldering he is among the best in Germany with his ascent of an 8c boulder. In addition to climbing on rock, he is passionate about competing at informal events.

**Thomas Dauser** grew up in the Altmühltal area in the southern part of the Franconian region. Climbing 9a and bouldering 8b, he is within the range of higher-elite-level climbers.

The following Table (Table 1) aggregates the standardised information as recommended by the IRCRA, as provided by the participants.

### 3.2. Influencing Factors on Personal Development as Climbers

The parents’ role is a strong influencing factor. Three participants started climbing with their parents at a reasonably young age (<10 years). The other three also had the full support of their parents, even though the parents did not climb themselves. All stated the importance of parental support. In particular, one participant stated his father had a major impact on the way he mentally approaches climbing. His father simply wanted to try every route at the cliff—when he failed the ascent, he would not bother to try that line again, but instead, immediately try the next line. The participant thought his relaxed mental relationship with climbing had its roots in his father’s behaviour. In addition to his parents’ influence, the participants described the presence of friends, competitors, and training partners as formative in their development as climbers. Moreover, five of the six participants mentioned their intrinsic motivation as an important factor. The intrinsic motivation, or “fire”, was described as one of the key elements of their personal development.


*I have a kind of fire in me, which somehow only started to burn through this climbing. (…) I always want to challenge myself, personally as well. And I always push myself to the limit, and to my emotional limit eventually, and that is my motor.*


For some participants, climbing felt more fulfilling than anything else they had experienced before, and was the first medium where they perceived themselves as outstandingly good. Furthermore, climbing gave them something upon which they could grow. All participants continued climbing for themselves, and kept training and climbing intensively without the need for external motivation.

### 3.3. Behaviour in Climbing—The Concept of Balanced Behaviour

Through the process of data analysis, three categories of social behaviour in the context of climbing were derived: negative egoistic behaviour, positive supportive behaviour, and balanced behaviour. The participants also outlined their perception of the influence of behaviour on climbing performance. Furthermore, similarities in social behaviour in school and in the climber’s self-perceived social role were found.

#### 3.3.1. Negative and Egoistic Behaviour

All participants gave examples of how their perception of a successful, good and satisfying climbing day had changed over time. In the past, they behaved more egoistically, prioritising their own climbing attempts, and did not care very extensively about the needs of their climbing partners.


*I mean, as a kid, or as a teenager, I just wanted to climb as much as I can, without consideration of losses. And I maybe did not care then that others get to climb less, because I smack in fifteen rapid fire attempts. It was rather like…I was too egoistic to say, okay, I draw the short straw, so another one gets to climb more.*


This changed over time, catalysed by specific events (crucibles), towards more supportive behaviour. The following example displays the most common scenario of crucibles in the present study.


*(…) [The memory] that has stayed with me the most was in 2016, I was just about to begin my studies. And I really wanted to climb [that specific line] before I started my studies, because I knew that it would be extremely difficult to travel much. [I was] just in a very tense situation in many respects, I think. And I just had a day when I was at the [Cliff] with my sister and her boyfriend, I think, and two others, and I noticed that I was just radiating an unbelievable dissatisfaction, and I was just beating myself up about why I couldn’t just climb it now, that I had no interest at all in the climbing itself, but only in the goal of having climbed the [route], and to have done it, and I just noticed that… I just noticed that I didn’t want to be like that, that I probably ruined the day for others, and that I don’t want that from myself. And that I’m obviously a worse climber when I’m like that, and that [this] actually only has disadvantages. And then I decided to stop [trying the line] and wait for the next spring, although… I don’t know, it was still the beginning of October, something like that, where there would have been a lot of time [and good conditions], but I realised that I was becoming a person I didn’t want to be.*


Possibly, this shift is inherent in every climber’s development.


*And this is where I can see the development, that everybody goes through in his younger years. When climbing plays such an important role, that it is used as benchmark for everything.*


Egoistic behaviour in the context of climbing is multifaceted. When rock climbing, it can be displayed by consciously deciding to attempt a climb at a time and place that suits oneself best, but is not ideal for the climbing partner.


*So let me put it this way, I used to be a lot more like this, and chose people, to get the best for myself out of it, to get done as much as I can and so on, and this has changed a bit.*



*(…) my route and I had the rockclimbing day, and there was not much else [social interaction].*


Another negative behaviour was remaining in a bad mood after a failed attempt or behaving stressed and rejective, in a way that also affected the climbing partner’s mood and, therefore, their performance. This appears to be common in climbing. Every participant was able to outline examples, either from their own behaviour or from someone else’s behaviour.


*(…) everybody else there knows too, that I am trying a difficult project, and they know, that I can be quite obnoxious sometimes, and pretty annoyed, and that I am very egoistic in that moment.*


This type of frustration regarding self-performance also often took away the joy of the climbing day itself, by linking joy only to success.


*I observe that people are just really ambitious about the grades, and that this takes a lot of fun out of it.*


One participant stated that he had little respect for climbing grades, and just kept trying routes that were above his current limit, until he pushed beyond his limit und eventually could climb the route. Conversely, another participant stated that it was of great importance not to try routes above his limit, thus, he could climb a significantly higher number of different lines, had to adapt to a greater variety of moves and held his motivation high due to frequent success. The absence of success may lead to repeated complaints about bad conditions and shape, negatively affecting the atmosphere at the climbing venue.


*Well, in general, what I don’t really like is the bad mood at the cliff. When someone is frustrated because of something, even if it’s absolutely clear that he or she will never get up the climb anyway, and then is always frustrated, and then always complains that yes, this is the very hardest 8a+ there is, and that’s not even true!*


Relationship tensions could also be transferred into the climbing session, further prohibiting the climber from being fully focused on the attempt.


*(…) it was often stressful for me because relationship stress meets climbing ambitions.*


Another negative impact was the presence of competitive and jealous behaviour, which can be seen in climbing even outside the context of competitions. Climbers would observe the attempt of a preceding climber on the same line they wanted to attempt; some climbers would be pleased to see the other climber fail. This took away the pressure, opened the opportunity to show off, and “prove” themselves to be a better climber than the one who failed.


*But with people who might be competitors somewhere, who now think, wow, he took a fall now, great, I’m really happy now, because he’s crapped out up there, I don’t have any more conversations with them, no.*



*I have the feeling that there are some people for whom it’s almost the other way round. They feel better when they know that other people are doing badly, because then they know, okay, cool, now I can take full advantage of it, so to speak, so the really competitive type [of a climber] (…).*


Moreover, disrespectful behaviour when climbing is becoming more common, due to the increasing popularity of sport climbing and the resulting increase in the number of climbing facilities. This has led to more people becoming familiar with and trying various forms of climbing, including climbing outdoors. Behaving disrespectfully in the context of climbing can be displayed as playing loud music, not cleaning holds, and generally not having an awareness for the needs of others.


*(…) and I would like to see respect from person to person growing, also on the rock outside and also in the gym. The respect. Because somehow… human respect is somehow gone, I have the feeling.*


A climber may then react by distancing themselves from disrespectful behaviours by choosing climbing locations where they are likely to be alone.

In the setting of official climbing competitions, the circumstances are different. When and where to climb, and in which order, is predetermined for every athlete. In this setting, different social interaction behaviours were limited. These social interactions could not be avoided since all athletes gather together in the same warm-up and isolation zones. In these scenarios, egoistic, self-centred behaviour appeared to be common.


*I always go into competitions, unless it’s a World Cup, with the feeling that I want to be as good as I somehow can be on that day. And I am very focused on myself, definitely. So, there is little room for me for any kind of support for someone else. On the other hand, the others are often like that too. At least, that’s how it often was, that there was simply nothing… nothing from the others either, of course. That’s also something that I do not necessarily like about competitions.*



*And then you could really feel that there was a sense of competitiveness in the air, and [among the climbers] there was a very striking competitive pressure somehow, they really looked at each other’s fingers to see how the skin is, looked at what people were eating before finals and before the quali route. It was massive.*


A specific negative behaviour in the context of competitions was to connect the personal mood directly to the outcome of the competition, and to only accept the accomplishment of a self-set goal, but not being fully satisfied by reaching it. In the worst case, this behaviour negatively affected the mood of the social surrounding.


*(…) then for me there was really only either I win, and that’s good, but that’s also only “good”, that’s just standard then, or I lose…*



*But I was just so upset that I wasn’t going to win this competition, that I saw again, that in this case my ex-girlfriend simply suffered from it, that she had to endure it. It wasn’t that I… that I was angry with her, but she was extremely impacted by my feeling of dissatisfaction.*


#### 3.3.2. Positive and Supportive Behaviour

The opposite of these negative behaviours is positive, supportive, and selfless behaviour. In the context of climbing, these were all behaviours that increased the likelihood for their climbing partners to have a fulfilling climbing session. This started with conscious acts of encouragement, of setting a good tone for the day and putting minor conflicts and tensions aside. Reducing pressure is one specific form of supportive behaviour that was common among the participants’ experiences. For instance, it could be achieved by assuring the climbing partner that it was possible to try their desired route as often as they wanted.


*So whenever I see someone getting close to [a successful ascent], I always tried to give them the feeling, okay, we can come here again.*


Sometimes though, the right amount of mental pressure can also be helpful. This pressure can be provided by the climbing partner and can be considered as another form of supportive behaviour, which particularly requires empathy. Moreover, it is supportive to shout empowering phrases while the climbing partner is attempting their route. It is important, to find the right timing and the right words, because there are personal differences of what is perceived as appropriate for the situation. Some also may find any sort of verbal encouragement uncomfortable while in a serious climbing attempt.


*(…) everyone kind of shouts Allez! Allez! And press! Come on! Go! Knüppel! Blah, blah, and I don’t hear many people. I can hear certain people, but not all of them. So, they can all shout as they want, but I don’t hear them all.*



*Many people say right away, hey, please don’t say anything. I always ask the climber now, if I don’t know him that well anyway, then I ask him if he needs a push somewhere.*


Moreover, the participants began to realise that lower-level climbers could also train hard and could invest time end effort into climbing, and that they deserved support as much as a higher-elite-level climber.


*(…) the world of others, so to speak, is also relevant, it has grown, in my eyes. I have friends, who I can watch climbing, they really put their heart and soul into it, and really want to get forward, and do their stuff well on their level! I started to see that too. At those days I never saw that someone who wanted to climb a 6 or something would then somehow try to do a pull-up. That was not visible to me at all! And meanwhile it has all become highly transparent for me.*


#### 3.3.3. Egoism and Support—Finding the Right Balance

All participants stated that over time, good mood and atmosphere in the climbing session became more important to them, and climbing success less important. This process was either unconscious, or was catalysed by specific events called crucibles [18]. Two of the six participants spontaneously described their own experience of crucibles. These crucibles were of major importance to the behavioural development of those participants, and moments or phases of deep frustration because of repeated failure to achieve a specific goal. In those moments, they would worsen their own, and their companions’, mood, performance, and atmosphere, on a scale that was not appropriate to the importance of the achievement, especially when considering what is personally valuable across the whole lifespan. During these times, the participants deeply questioned their behaviour, and had to adapt to achieve their goal in a satisfying way. Both stated that this process made them the climbers they are now, and had a huge influence on their whole personality. In addition, four of the participants could point to one or multiple persons that acted as their mentors for behaviour, in the context of climbing. These persons also showed support during their harder times in life, and influenced the climbers in several parts of their lives.

In all cases, these formative experiences involved some sort of failure on a climbing project or in a competition, that led to immense self-made pressure, frustration, and loss of joy, dragging down their own mood and the mood of their companions. The participants felt that it was necessary to be distanced from the project for a while, to achieve an altered perspective on inner motivation, and on what really matters personally.


*(…) I had a bouldering project where I was fully challenged, physically, emotionally, really deeply, that was really hard. Then I was injured, then I regenerated in the boulder, then I had to be distanced, I also needed it, consciously, and then I really… that really shaped me, if you can say that, you grow as a person in experiences. That’s what I experienced for the first time, that I felt grown up when I did the boulder after six years. (…) I developed to what I am now. And I learned a lot of things [within myself and things] that are important to me, and that would never have been the case without this bouldering project. (…) And that was the most memorable cut in my life.*


A balance between serving and being served, between supportive and egoistic behaviour, had to be found to set the base for a fulfilling climbing session for all people involved.


*(…) I think in the long run it’s always healthier if no one has the feeling that they have to give in. And that’s what I tried to, yes, arrange somehow.*


The participants achieved this balance through a variety of changes in behaviour, ranging from giving in and not trying another route, but instead happily letting climbing partners have additional attempts, to learning techniques for consciously controlling bad mood and frustration, and learning how to better discuss the timing of attempts. They also learned which climbing partner that balance could best be achieved with, and developed a sensitivity for the other’s needs and efforts.


*We supported each other completely, without talking much actually.*



*So, what I’ve been doing more of lately is, that I see, that others have their needs too.*


It also appears that the type of social role a climber adopts, whether predominantly supportive or egoistic, also depends on how close someone in the group is to a successful and personally meaningful ascent. The closer the climber was to success, the more priority their attempts were given within the group of people involved. The others would tend to act more supportively, and the climber performing the attempts would tend to behave more egoistically.


*But it also depends on what my role is. [If I am going to try the] project that I’ve been working on for 5 years [in the same session], then it would probably be different. Then of course I would also try to spread a good vibe, but then I would be much more tense.*



*I mean, sometimes it also depends on the one who is maybe closer [to the ascent]… then [this is where] you have the priority, that maybe he does it somehow.*


#### 3.3.4. Enhancing Climbing Performance through Balanced Behaviour

Finding the balance between egoistic and supportive behaviour was perceived by all participants as an improvement of their climbing performance. Feeling that the circumstances for their climbing partners were not ideal and that they could not enjoy the session had a negative influence on the thoughts of the participants, prohibiting them from fully focusing on their goal.


*No, that has always influenced me. So, when I had the feeling that other people somehow weren’t up for it because they didn’t get to climb, or because they were hanging out at a cliff where there was nothing [to do] for them, then that always had a massive influence on me.*


The negative behaviour of being frustrated and rejective also had an influence on the climbing partners’ mood, which worsened the mood of the participants even more, in a downward spiral. This led to the participants’ realisation that being supportive to a certain extent, where the balance was right, enhanced their own climbing performance. The participants’ own performance was perceived as better when they were also observant of others’ needs, than when the participants solely focused on their own needs and neglected the needs of others.


*But conversely, if climbing has such a massive influence on your mood that it’s not benefiting the performance and social environment and general mood… [I] then realised that it’s actually mega inefficient to work like that, because it’s simply not beneficial for climbing performance at all.*


The occupation of mental focus on thoughts about the well-being of others can also be observed in reverse: when the climbing partner is in a bad mood, complains about conditions and own performance, and the atmosphere is perceived as depressing as a result. In those cases, all participants also felt a decrease in their own climbing performance, because their thoughts were occupied by social tensions. For some participants, the component of social interaction and its risk of decreasing performance sometimes even led to avoidance of social interactions.


*So, what I’ve noticed in recent years, and what is a total motivation-killer for me, is when I go climbing with people and they don’t feel it and just complain about everything. So, when people around me are already so unmotivated and somehow don’t feel it, then that really gets me also down. And I don’t have any fun anymore.*



*With some people it just doesn’t work out. And then I can’t try things in the project the way I would if I were with other people. And then it’s no use trying the project, because then I don’t make any progress if I’m constantly thinking about something else.*


Choosing only a few close climbing partners, whose behaviour was familiar, appeared to be a solution for some of the participants. For the others, it was the opposite: going climbing with many different partners of different climbing-grade levels, to minimise unconscious competitive pressure.

#### 3.3.5. Social Behaviour in School

All participants recalled that they were never the centre of social interaction in school. Four of the six participants noticed themselves as being the “special ones” in school, with a few close friends. Their relationship with the rest of the class was neutral. However, they did not wish for more, they just did not have the urge to integrate themselves fully into social life and were happy to focus on climbing.


*We were busy with ourselves in school somehow and didn’t care about other people. What they did didn’t matter to me. They did their stuff, which was fine, we accepted that, but it wasn’t really important to us.*


The remaining two participants were clearly divided between social groups in climbing and social groups in school, and were well integrated in both. They felt that the balance between the integration in climbing and non-climbing groups was just right for them.

## 4. Discussion

This study’s interviews revealed the influence of social behaviour on climbing performance. An optimal balance of egoistic and supportive behaviour appears to be crucial for attaining maximised performance in climbing. This observation can be represented by the *concept of balanced behaviour* (Figure 1).

The scale of social behaviour in climbing is loaded with supportive, positive, and empathic behaviour on one side, and with egoistic, negative, and selfish behaviour on the other. If the positive side is loaded too heavily, it represents the climber deferring their own needs for the sake of others’ performance or joy. The overly supportive climber might leave all the good conditions to his climbing partner, while settling with poor conditions. Conditions can be of mental (amount of stress), physical (time of the day, muscle activation) or environmental nature (light, temperature, humidity). Supportive climbers may also never reach their full potential in climbing because they cannot concentrate on pursuing their own climbing goals. On the other hand, an overloaded egoistic side on the scale of social behaviour also decreases performance, due to the mental backlash of the climbing partners’ bad mood [5]. The decreased performance leads to dissatisfaction, which further worsens the mood. Figure 2 summarises the negative effects of unbalanced behaviour. As described, predominantly egoistic behaviour leads to what can be called a *downward spiral of mood*.

Consequently, the scale’s bar should be kept horizontal, to ensure that social interactions are no hinderance towards a fulfilling climbing day for each member of the group. All participants of this study’s interviews experienced a development over time, from egoistic to balanced behaviour, from an uneven to an even bar. The whole process of overcoming one’s own egoistic behaviour, towards a balanced behaviour, appears to facilitate personal development on the way to higher-elite-level climbing.

A similar effect may be observable in other (individual) sports. Again, the influence of social behaviour on sport performance has surprisingly received little attention. Only a recent topic review on the influence of *empathy* on physical performance [21] was found in the literature. The study suggested that “observing sad photos or videos or fatiguing exercise can adversely affect subsequent performance”. Empathy also plays a role when feeling the needs of others in order to act supportive.

All participants described a development from predominantly egoistic to more balanced behaviour. These perceived changes in behaviour through sporting activities are accordance with recent studies, where sport was observed as beneficial for social behaviour in general [22,23,24]. The participants’ former predominantly egoistic behaviour in climbing may be rooted in their tendency towards self-centred behaviour at the time when they still went to school. This result is in accordance with Brandauer’s [14] conclusion, that the male sport climbers of his study describe themselves as “introverted, less contact seeking, and reserved”.

### 4.1. Balancing the Bar

The ideal area of the bar on the scale of balanced behaviour is horizontal. According to the statements of this study’s participants, most of the younger climbers were more on the egoistic side in their average behaviour. Further research is required to validate this theory. If a person recognises himself with a tendency to be on one side of the bar, the aim should be to balance that bar using appropriate counterweights. The mass of these weights varies interpersonally. One person would need heavier counterweights than another, as displayed by the participants’ examples. One participant perceived himself as naturally relaxed, and stated that his development towards balanced behaviour did not involve drastic changes. In this case, the weight is light on both sides. Another participant uses heavy weights to balance behaviour. He allocates time into phases. In one phase, he works intensively on his project and makes it clear to his group that he will be more egoistic during this time (heavy weight on the egoistic side). In the following phase, he steps back and supports wherever he sees the possibility (heavy weight on the supportive side). Even if egoistic behaviour dominates in one phase, the bar remains balanced, as long as the climber clearly communicates his behaviour with the people involved and acquires their agreement. In this case he just postpones his supportive behaviour into the following phase. Therefore, in the total time period, the bar remains balanced. Without clear communication and agreement, the scale would be off balance, because others may be negatively affected by the climber’s behaviour. The downward spiral of mood may be triggered. Therefore, beside discovering on what side of the scale one tends to be, it is also necessary to consciously and individually find the right-sized weight to counterbalance the current behaviour. The form of (behaviour) weights on either the egoistic or supportive side appear to be highly individual, and require good communication with everybody involved.

### 4.2. Extreme Misbalance–Fatal Supportive Behaviour

More than in sport climbing, success in mountaineering is dependent on combined team effort. Success may lay in returning home alive. In extreme “cutting the rope” situations, one must decide between saving one’s own life or the high risk of dying in the attempt to save the partner’s life [25]. A decision is made between supportive behaviour, which would probably lead to death, and “egoistic” behaviour, to save one’s own life. One example of where the decision fell on the supportive side is the tragedy of 1996, where more men and women than ever before died in their attempt to reach the top of Mount Everest [26]. Robert “Rob” Hall, a world-renown high altitude mountaineering guide also lost his life on that peak that year. He was waiting on the summit of Mount Everest for his client, Doug Hansen, to join him. While waiting, he let the time point pass, that he usually sets, where it is absolutely necessary to turn around for descent, regardless of whether the summit has been reached [26]. He waited for two more hours for Doug to arrive. When Doug finally reached the summit, Rob was too fatigued for the descent. In the following twenty-four hours, a series of incidents (failed communication, cold and strong winds, and illness of other mountain guides) hindered both mountaineers from reaching the nearest camp. Doug eventually fell off a narrow crest near the summit. Rob later died from the freezing cold after exchanging his last words with his wife via radio. This is an extreme example of supportive behaviour. The reason for this could lay in the event’s foreplay: one year before, Rob had to lead Doug to descend when he was within 100 m of the summit of Mount Everest; later, he called Doug several times and eventually convinced him to try another attempt [26]. This story shows that the concept of balanced behaviour may be also applicable to mountaineering and other sports than climbing.

### 4.3. Limitations

Despite its world class level, the small sample (*n* = 6) must be seen as a limitation. The present findings can only be used as novel perspectives to build upon, and as directions for larger sample studies in the field of social behaviour in sport climbing. As the first of its kind, this study cannot present general conclusions. It must also be noted that interviews only display the self-perception of the participant. For each participant, other close persons would be needed to validate the self-perception, especially for the changes in behaviour over time. Balanced behaviour is intuitively perceived as behaviour worth pursuing, so one would tend to describe their own behaviour accordingly. Nevertheless, this does not affect the validity of the participants’ statements regarding their personal background of development.

Furthermore, this study included only German male athletes. It remains open as to whether other cultures have developed different patterns of social behaviour in the context of climbing, and if gender specific differences exist.

Additionally, the disclosure of the athletes’ names and brief history might influence the interpretation of the results, as the opinion of an Olympian might subjectively weigh more than the opinion of an anonymous climber. Still, the data are worth disclosing, for better comparison of this study’s participants with the participants of other studies, and for creating a framework upon which the quotes and results can be interpreted.

## 5. Conclusions

This study aimed to identify patterns of social behaviour in higher-elite-level climbers, and to examine their influence on climbing performance. The study was performed with explorative, semi-structured in-depth interviews. All six participants reported a development from egoistic behaviour towards a more supportive behaviour, where they ultimately aimed for a balance between those extremes of behaviour. This can be called the concept of balanced behaviour. The behaviour-changing development was a time-intensive process. For some participants, specific critical events (crucibles) acted as catalysts. Finding the right balance between supportive and egoistic behaviour appears to be an important ability as a higher-elite-level climber. Therefore, coaches and athletes should be aware of this performance-influencing factor and may increase their awareness for social behaviour. It is considered as reasonable to aim for balanced behaviour. Further investigations can build upon these first insights into higher elite climbers’ behaviours, and researchers are encouraged to seek validation of the concept of balanced behaviour by using large sample sizes and quantitative methods. Differences between diverse cultures and gender should also be included in further research. To put it in a nutshell: climbers are advised to have respect for the needs of others and to aim for a balance of supportive and egoistic behaviour, to optimise their personal climbing performance.

## Figures and Tables

**Figure 1 ijerph-19-09703-f001:**
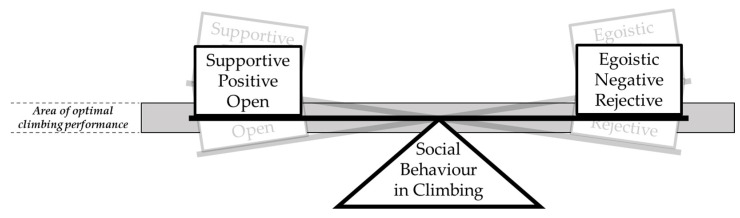
Concept of balanced behaviour. The bar has to be within the horizontal area to provide a setting for optimal climbing performance.

**Figure 2 ijerph-19-09703-f002:**
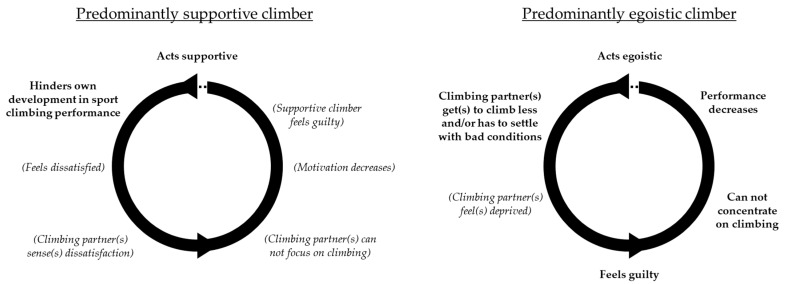
Negative effects of unbalanced behaviour. Scenarios for predominantly supportive or egoistic behaviour based on the participants’ experiences. The effects can be repetitive and self-amplifying. In most cases, one full circle describes one climbing session.

**Table 1 ijerph-19-09703-t001:** Participant profiles as suggested by IRCRA (2015).

	Alexander Megos	Christian Münch	Moritz Welt	Philipp Hrozek	Stephan Vogt	Thomas Dauser
Age (years)	28	33	20	40	27	34
Climbing experience (years)	21	21	16	25	11	24
Competition experience	International	National	National	International	International	National
Main discipline	Lead	Lead	Lead	Bouldering	Lead	Lead
Ratio lead/bouldering	20/80	70/30 (est. *)	75/25	Past: 10/90. Currently: 80/20	Outdoor: 90/10 (est. *). Indoor: 5/95	75/25
Ratio indoor/outdoor	60/40	10/90	30/70	75/25	67/33 (est. *)	33/67
Training and climbing (hours/week)	20–30	20–25	20–25	Up to 28	3–20 (est. *)	15
Max. lead (max. 9c)	9b+	9a	9a+	9a	9a	9a
Max. boulder (max. 9a)	8c	8b	8c	8b+	8c	8b

Legend. est: *: when a participant was not able to point out a specific ratio, the author estimated what was most likely meant by the participant’s answer and transferred it into metrical comparable data.

## Data Availability

The data of the study are available on reasonable request.

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
