# Peer review of "Balancing the Bar–Influence of Social Behaviour on Sport Climbing Performance"

_ijerph, 2022, doi:10.3390/ijerph19159703_

Round 1

Reviewer 1 Report

First of all, I would like to share the need to carry out works like the one you present. They are necessary for the advancement of science in the field they study. Thank you for allowing me to review this interesting study. In general, the proposed work generates a very interesting point of discussion. I believe that this study has provided novel findings in this area, allowing readers to think more deeply about what is happening around the sport of climbing based on the performance of their athletes. The purpose of the manuscript is clear and consistent. The study has been an interesting read, it is necessary to know the reality of the sector on which the work emphasizes. I find it really interesting. It is a novel study, which is positive on the one hand, but the lack of scientific evidence on the sector is highly complex. The abstract includes the necessary elements: background with the purpose (objective) of the study, methods, results, main conclusions without exaggerating them.

In the introduction, sufficient ordered references of the publications considered key, with significant and sufficient evidence, are indicated. But you should focus only on the specific types of climbing in the study. Likewise, reasons that justify the importance in a broad context and the current state of the investigated topic are highlighted. The study is clearly defined and indicates the intent and meaning of the work. The objective to be tested in the study is recorded. The text is understandable and makes clear the main objective of the work and the main conclusions.

In relation to the material and methods, say that the study is described in detail. However, the main limitation of the work is the low number of participants in the study. Although it is true to obtain such a specific sample of high-level athletes, it is highly complex, and as a reviewer I am aware of this. However, it is the main limitation and it must be left reflected in the document. In addition, establishing general conclusions based on these 6 athletes is highly difficult, so the data must be treated with caution and must be left reflected.

There is little scientific evidence of the sector they work in, so recent work carried out in the same field of study as the work submitted for evaluation should be included. Reference should be made to works such as Morenas, J., Luis del Campo, V., López-García, S., & Flores, L. (2021). Influence of On-Sight and Flash Climbing Styles on Advanced Climbers’ Route Completion for Bouldering. International Journal of Environmental Research and Public Health, 18(23), 12594.

With regard to the history of the different climbers, it is highly complex to publish this data, so it is recommended that the names be coded and be anonymous if necessary, unless the consent specifically gives permission to do so. spread, but keep it in mind.

Perhaps disclosing your data can influence the results of the interviews, reflect it as a limitation.

Reviewer 2 Report

I am glad I had a chance of reviewing this paper, and I hope my comments will help to enhance and broaden the perspective. The study is innovative and addresses important information on the area of social interactions during climbing sessions well-being. Overall, this is a well written article. Even though, the manuscripts present some flaws that must be considered. I recommend its publication after minor changes. Some additional details could be added to the method section, below you will find my specific recommendations.

1)the authors would have to describe the unique contribution of this study to the current literature. A more detailed description of the methodology is required.

2)Abstract: The summary should follow the style of structured summaries (background, methods, results, and conclusions).

3)Materials and methods/This section needs to be described a little more, such as what the inclusion and exclusion criteria were.

4)Discussions/Point out the strengths and limitations of your study.

5)The general objective and specific objectives should appear at the end of the introduction.

6)The sample is very small for this type of study that should pay careful attention to its inference results, and should be limited in the article.

Round 2

Reviewer 1 Report

Dear Sir/Madam,

I appreciate your reply to my review. The changes you ha e made have noticeably imprived the quañity of your manuscript.

Regards